# Identifying genetic modifiers of age-associated penetrance in X-linked dystonia-parkinsonism

Björn-Hergen Laabs [1], Christine Klein [2✉], Jelena Pozojevic[2,3], Aloysius Domingo[2,4], Norbert Brüggemann[2,5], Karen Grütz [2], Raymond L. Rosales[6,7], Roland Dominic Jamora [8], Gerard Saranza[8], Cid Czarina E. Diesta[9], Michael Wittig[10,11], Susen Schaake[2], Marija Dulovic-Mahlow[2], Jana Quismundo[2], Pia Otto[2], Patrick Acuna[4], Criscely Go[12], Nutan Sharma[4], Trisha Multhaupt-Buell[4], Ulrich Müller[13], Henrike Hanssen[2,5], Fabian Kilpert[14], Andre Franke [10,11], Arndt Rolfs[15,16], Peter Bauer[15], Valerija Dobričić[2,17], Katja Lohmann[2], Laurie J. Ozelius[4], Frank J. Kaiser[3,14,18], Inke R. König [1✉] & Ana Westenberger [2✉]

X-linked dystonia-parkinsonism is a neurodegenerative disorder caused by a founder retro-transposon insertion, in which a polymorphic hexanucleotide repeat accounts for ~50% of age at onset variability. Employing a genome-wide association study to identify additional factors modifying age at onset, we establish that three independent loci are significantly associated with age at onset ($p < 5 \times 10^{-8}$). The lead single nucleotide polymorphisms collectively account for 25.6% of the remaining variance not explained by the hexanucleotide repeat and 13.0% of the overall variance in age at onset in X-linked dystonia-parkinsonism with the protective alleles delaying disease onset by seven years. These regions harbor or lie adjacent to *MSH3* and *PMS2*, the genes that were recently implicated in modifying age at onset in Huntington's disease, likely through a common pathway influencing repeat instability. Our work indicates the existence of three modifiers of age at onset in X-linked dystonia-parkinsonism that likely affect the DNA mismatch repair pathway.

[1] Institute of Medical Biometry and Statistics, University of Lübeck, University Hospital Schleswig-Holstein, Lübeck, Germany. [2] Institute of Neurogenetics, University of Lübeck, Lübeck, Germany. [3] Section for Functional Genetics, Institute for Human Genetics, University of Lübeck, Lübeck, Germany. [4] The Collaborative Center for X-linked Dystonia Parkinsonism, Department of Neurology, Massachusetts General Hospital, Charlestown, MA, USA. [5] Department of Neurology, University of Lübeck, Lübeck, Germany. [6] Department of Neurology, University of Santo Tomas Hospital, Manila, Philippines. [7] Department of Psychiatry, University of Santo Tomas Hospital, Manila, Philippines. [8] Department of Neurosciences, College of Medicine - Philippine General Hospital, University of the Philippines, Manila, Philippines. [9] Department of Neurosciences, Movement Disorders Clinic, Makati Medical Center, Makati City, Philippines. [10] Institute of Clinical Molecular Biology, Christian-Albrechts-University of Kiel, Kiel, Germany. [11] University Hospital Schleswig-Holstein (UKSH), Kiel, Germany. [12] Department of Neurology, Jose Reyes Memorial Medical Center, Quezon City, Philippines. [13] Institut für Humangenetik, Justus-Liebig-Universität, Giessen, Germany. [14] Institute of Human Genetics, University Hospital Essen and University of Duisburg-Essen, Duisburg-Essen, Germany. [15] CENTOGENE GmbH, Rostock, Germany. [16] Medical Faculty, University of Rostock, Rostock, Germany. [17] Lübeck Interdisciplinary Platform for Genome Analytics, University of Lübeck, Lübeck, Germany. [18] EZSE - Essener Zentrum für Seltene Erkrankungen, Universitätsmedizin Essen, Essen, Germany. ✉email: christine.klein@neuro.uni-luebeck.de; inke.koenig@uni-luebeck.de; ana.westenberger@neuro.uni-luebeck.de

While recent years have seen unparalleled advances in identifying disease genes for monogenic neurogenetic conditions, a mostly unresolved challenge of personalized medicine remains the question as to whether a mutation carrier will manifest the disease and, if so, when and to what extent. Although elucidating the factors that contribute to this variability in (age-related) penetrance and disease expressivity has a high imperative and translational potential, this is typically impeded by the occurrence of private mutations with different effects, variable genetic and environmental background of patients, and small sample sizes. In contrast, X-linked dystonia-parkinsonism (XDP, DYT/PARK-*TAF1*, OMIM #314250) is a rare disease where patients share a founder mutation and come from the same locale (Panay Island, Philippines). An active group of local clinicians and researchers have made it possible to accumulate relatively large sample sizes for a disease with a global prevalence of <1:1,000,000, making XDP an ideal model to study genetic modifiers of disease expression[1].

XDP is an adult-onset movement disorder characterized by striatal neurodegeneration and inherited in an X-linked recessive manner, thus predominantly affecting men. All patients carry the same disease-causing mutation, a ~2.6-kb SINE-VNTR-Alu (*SVA*)-type retrotransposon insertion in intron 32 of the *TAF1* gene[2,3]. The XDP phenotype, already characterized by the presence of two different movement disorders (dystonia and parkinsonism), shows extraordinary variability in age and site of disease onset, as well as in the type of initial signs and rate of progression. Recently, we showed that the length of a polymorphic $(CCCTCT)_n$ repeat within this *SVA* retrotransposon insertion correlates inversely with age at onset (AAO)[4,5] and *TAF1* expression, as well as positively with disease severity and cognitive dysfunction[4]. The number of hexanucleotide repeats, however, accounts for ~50% of the AAO variability in the XDP patient population[4,5] and even less of the variable expressivity of XDP[4], suggesting additional modifiers.

In this work, we employ a genome-wide association study (GWAS) approach to search for additional factors influencing age-related penetrance of XDP. Our findings (i) confirm that genome-wide significant genetic modifiers can be identified in small but homogeneous samples of patients with monogenic disease; (ii) reveal a biologically plausible mechanism of disease modification in XDP; and (iii) pathophysiologically link XDP to Huntington's disease (HD, OMIM #143100), another repeat expansion disorder with severe striatal neurodegeneration as an overlapping feature of both diseases.

## Results

**Genome-wide association study.** Using DNA from 353 genetically confirmed XDP patients (Supplementary Table 1), we performed a GWAS in search of genetic modifiers of AAO in XDP. The average AAO (±standard deviation (SD)) of our patient cohort was 41.8 ± 8.4 (range: 21–67) years. The mean (±SD) detected $(CCCTCT)_n$ repeat number was 41.6 ± 4.1 (range: 30–55). Repeat numbers showed an inverse correlation with AAO (Pearson product-moment correlation: $r = -0.703$, $p < 2.2 \times 10^{-16}$), explaining 49.3% of the AAO difference in our 353 XDP patients ($R^2 = 0.493$). Association of AAO with repeat number was previously investigated in 328 of these 353 patients[4,5], and their combined analysis yielded an $R^2$ of 0.488.

In the screening step, we identified 93 candidate single-nucleotide polymorphisms (SNPs) to cover potential genetic modifiers influencing AAO (Fig. 1a, Supplementary Table 2). Of those, 60 SNPs are located within a 0.33 Mb region on chromosome 5 (rs173441-rs10075662; GRCh37 coordinates: 79893519-80222980) (Fig. 1b), and 33 SNPs are situated on

chromosome 7 within a 0.04 Mb area (rs17136307-rs660547; GRCh37 coordinates: 6064799-6106731) (Fig. 1c).

Out of the 60 SNPs within the candidate region on chromosome 5, 34 variants are located in the *MSH3* gene (Fig. 1b). A further 26 SNPs are located in the *DHFR* gene ($n = 4$) and in the intergenic regions 5′ from *DHFR* and 3′ from *MSH3* ($n = 22$) (Supplementary Table 2). All of the SNPs in *DHFR* and 33 of 34 *MSH3* variants are non-coding (intronic). The only coding *MSH3* SNP, rs1650697, p.Ile79Val, is in exon 1. The SNP with the most significant association with AAO in our GWAS ($p = 1.1 \times 10^{-12}$) is rs245013 located in intron 12 of *MSH3* (ENST00000265081) and conferring an estimated mean decrease in AAO of 3.30 years per alternative allele (Fig. 2). Iterative conditional analyses revealed that some of the SNPs on chromosome 5 continue to be significantly associated with AAO, even if the regression was adjusted for this variant (rs245013) (Supplementary Fig. 1), indicating a second independently associated region within the chromosome 5 locus. This signal includes 6 variants (with $p < 5.0 \times 10^{-4}$ after adjusting for rs245013) situated in a 0.06 Mb area spanning intron 20 of *MSH3* to intergenic region 3′ from *MSH3*. Among the 6 variants, rs33003 in *MSH3* intron 23 has the lowest $p$-value ($p = 7.5 \times 10^{-5}$) after adjustment and confers an estimated mean decrease in AAO of 3.19 years per alternative allele (Fig. 2).

Of the 33 SNPs within the associated region on chromosome 7, 22 are situated in the *EIF2AK1* and *ANKRD61* genes. Intron 11 of *EIF2AK1* harbors rs62456190 (ENST00000199389) (Fig. 1c), the variant with the lowest $p$-value ($1.3 \times 10^{-9}$), that confers an estimated mean increase in AAO of 3.82 years per alternative allele (Fig. 2). The remaining 11 variants are situated within the intergenic region 5′ from *EIF2AK1*. Only two SNPs on chromosome 7 are exonic variants (rs4724769, p.Lys18= in exon 1 and rs2302334, p.Met355Leu in exon 3, ENST00000409061), both in *ANKRD61*. Conditional analyses indicated a single significant region on chromosome 7, as no other SNP remained significant after adjustment for rs62456190 (data not shown). Of note, the genes on chromosome 7 closest to the region associated with AAO in XDP are *AIMP2* and *PMS2*, positioned 1.3/7.1 kb and 21.8/16.0 kb from the associated region/lead SNP (rs62456190), respectively.

When aggregating the $p$-values of single SNPs per gene, only *MSH3* was significantly associated with AAO ($p = 1.72 \times 10^{-12}$). We did not observe any genetic pathway to be significantly associated with AAO at the stringent significance threshold (data not shown).

In summary, the lead SNPs resulting from our GWAS and conditional analyses suggest the existence of three potential genetic modifiers of AAO in XDP, two within or immediately adjacent to *MSH3* and *DHFR*, and one within or immediately adjacent to *ANKRD61*, *EIF2AK1*, and *PMS2* (Fig. 1). The three most significant SNPs (one from each of the three regions) collectively account for 25.6% of the remaining variance (not explained by the hexanucleotide repeat) and 13.0% of the overall variance in AAO in XDP. Since the *SVA* repeat number is also a genetic factor, the variance explained by the combination of repeat number and the three independent loci can be interpreted as a heritability, leading to an overall heritability of 63% (where the repeat number accounts for 79% of the heritability and the three independent loci for the remaining 21%). Patients carrying the harmful or protective genotypes at all three SNPs are expected to have an AAO 7.9 years earlier or 7.0 years later, respectively, than predicted by the number of XDP-relevant hexanucleotide repeats alone.

**Analyses of variants in *MSH3* exon 1.** Using Sanger sequencing, we validated genotyping of the missense SNP closest to the strongest signal in our GWAS (*MSH3* exon 1: rs1650697, c.235A>G,

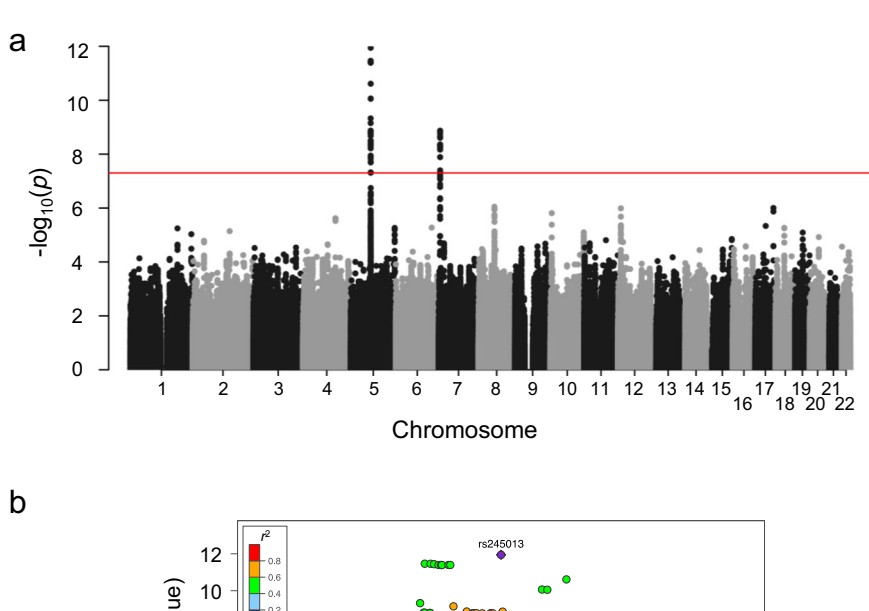

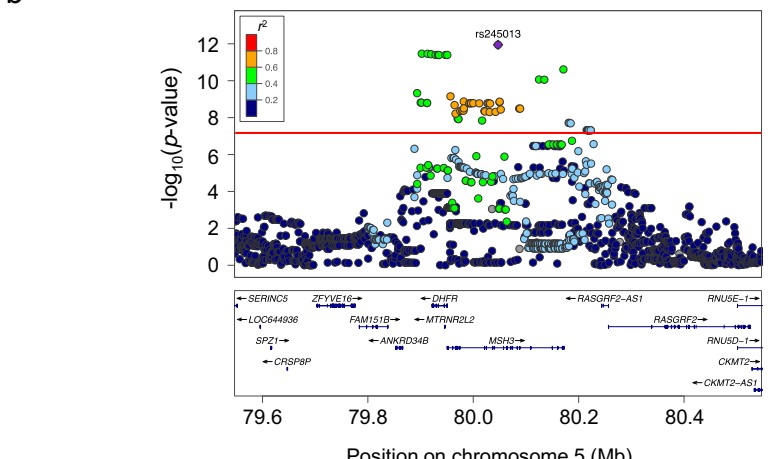

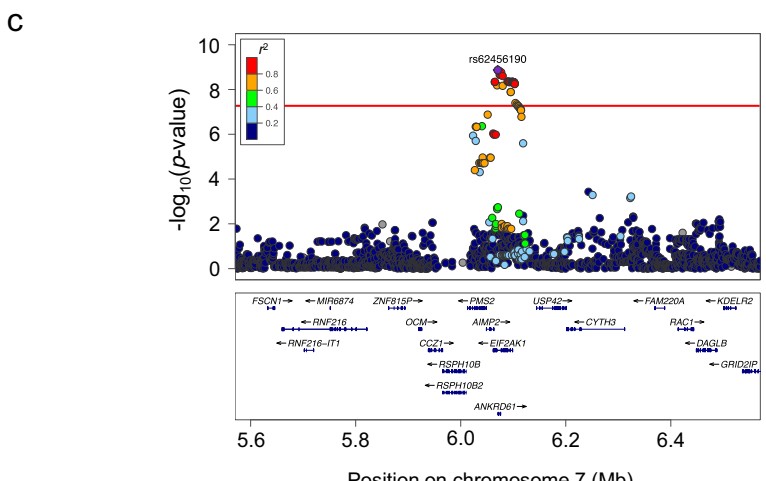

**Fig. 1 Genome-wide association study (GWAS) on age at onset (AAO) in X-linked dystonia-parkinsonism (XDP) patients. a** Manhattan plot of the XDP AAO GWAS. We estimated a series of linear regression models assuming SNPs with a *p*-value below $5 \times 10^{-8}$ to be significantly associated with AAO adjusted for repeat number. Using this approach, we identified 93 candidate SNPs to cover a potential genetic modifier influencing AAO. **b**, **c** Locus zoom plot of the GWAS data. The structure of linkage disequilibrium and –log10 (*p*-value) of the identified loci on chromosomes 5 and 7, respectively. Genomic coordinates are given according to GRCh37. Red lines: $p = 5 \times 10^{-8}$; Mb: megabase; each dot: $-\log10$ (*p*-value) of a SNP; $r^2$: the Pearson coefficient of correlation between any SNP in the graph and the SNP with the lowest *p*-value in that region. Source data are provided as a Source data file.

p.Ile79Val) in a subset of 285 XDP patients (Supplementary Table 1) with a validation rate of over 99%.

Sequence analysis of *MSH3* exon 1 revealed ten single-nucleotide and in-frame sequence length polymorphisms in close proximity of p.Ile79Val (Supplementary Table 3). We sequenced

exon 1 of *MSH3* in 95 additional XDP patients whose mean AAO and hexanucleotide repeat number did not differ from the GWAS cohort (Supplementary Table 3). After adjustment for *SVA* hexanucleotide repeat number, we calculated the impact on AAO of eight variants (including p.Ile79Val) in *MSH3* exon 1 with a

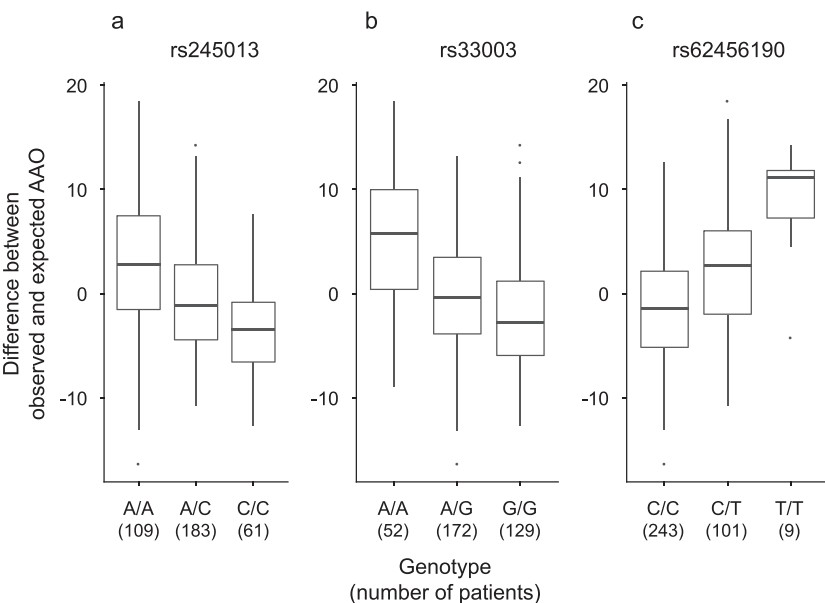

**Fig. 2 Relationship between age at onset (AAO) and the lead single-nucleotide polymorphisms (SNPs) in the three independent regions identified in our genome-wide association study (GWAS). a, b** Two independent putative genetic loci on chromosome 5, the alternative alleles of which were found to accelerate disease onset. **c** The alternative allele of the chromosome 7 signal delays AAO in our patients. Box-plot elements: center line: median; box limits: upper and lower quartiles; whiskers: 1.5× interquartile range; single points: outliers. Source data are provided as a Source data file.

MAF of >0.01 in 380 XDP patients in whom this exon was sequenced. Seven of these variants were associated with AAO with different directions of effect (Supplementary Table 3). Of note, the absence or presence of three in-frame sequence length polymorphisms [c.162_179del (p.Ala57_Ala62del), c.199_207del (p.Pro67_Pro69del), and c.181_189dup (p.Ala61_Pro63dup) the first two of which were always detected together; Supplementary Table 3] form alleles of three different sizes: (i) wild-type *MSH3*, (ii) a 27-nucleotide/9-amino-acid shorter, and (iii) a nine-nucleotide/3-amino-acid longer form previously described as 6a, 3a, and 7a, respectively (Fig. 3a)[6].

Interestingly, we found that the lengths of these three variants inversely correlate with AAO in XDP. When considering the residual AAO variability (difference between the observed AAO and the AAO expected based on the hexanucleotide repeat number), patients carrying the 7a allele showed the most pronounced mean reduction in AAO, while patients heterozygous for the 3a allele had the highest mean increase in AAO (Fig. 3b and c). Of note, in our cohort, only two patients were homozygous for the shortest (3a) allele, and their AAOs were considerably higher than predicted based on the hexanucleotide repeat number (expected AAO of 38 and 46 years vs. observed AAO of 55 and 64 years).

In addition to testing single variant effects, we also assessed the association between the six haplotypes we identified in our cohort (with a frequency of >1%) and AAO (Supplementary Table 4). The four most common haplotypes showed a strong association with AAO, with H1 and H3 having an increasing (+1.70 and +3.21 years, respectively) and H2 and H4 a decreasing (−1.94 and −3.70 years, respectively) effect on AAO.

**Effect of Huntington's disease polygenic risk score on age at onset in XDP.** We constructed a polygenic risk score (PRS) according to Gusella[7] to predict the AAO in XDP and we observed an increase of 1.30 years ($p = 1.06 \times 10^{-9}$) for every increase of one in the PRS. In total, the PRS explained about 10% of the remaining variance in AAO, while the three lead SNPs of our GWAS explained more then 25% of the remaining variance.

Overall, the PRS for HD was constructed based on a GWAS performed in observations from a different genetic population. Thus, the effect estimates are not necessarily transferable to a Filipino population.

**Expression quantitative trait locus query and *MSH3* and *PMS2* expression analysis.** Our query of the Genotype-Tissue Expression (GTEx) Portal[8] revealed that rs245013 is a significant expression quantitative trait locus (eQTL) for *MSH3* in multiple brain tissues including cortex (normalized effect size (NES) 0.32, $p = 3.6 \times 10^{-6}$), amygdala (NES: 047, $p = 6.0 \times 10^{-6}$), nucleus accumbens (basal ganglia) (NES: 0.29, $p = 1.3 \times 10^{-8}$), and putamen (NES: 3.1, $p = 3.1 \times 10^{-6}$) (Supplementary Data 1, Supplementary Fig. 2). The strongest signal within the second locus on chromosome 5 (rs33003) was not found in the GTEx Portal. The only coding *MSH3* SNP, rs1650697, is a significant single-tissue eQTL for *MSH3* in the cortex (NES: −0.45, $p = 2.3 \times 10^{-7}$) and nucleus accumbens (NES −0.34, $p = 2.3 \times 10^{-8}$) (Supplementary Fig. 2), and whole blood (NES −0.58, $p = 4.5 \times 10^{-63}$) (Supplementary Data 1). The lead SNP on chromosome 7 (rs62456190) is a significant eQTL for *PMS2* in coronary artery (NES: 0.42 $p = 7.1 \times 10^{-9}$), pancreas (NES: 0.40, $p = 8.4 \times 10^{-11}$), pituitary (NES: 0.36, $p = 3.3 \times 10^{-7}$, etc. (top associations based on NES, Supplementary Data 1).

Queries performed using the BrainSeq dataset[9] from the eQTL Catalog[10] as well as the United Kingdom Brain Expression Consortium (UKBEC) dataset did not identify significant eQTLs for *MSH3* (Supplementary Table 5, Supplementary Data 2)[11].

*MSH3* expression was not associated with any of the lead SNPs on chromosome 5 nor with variants in exon 1 of *MSH3* at a significance level of 0.05/7 = 0.0071 (Supplementary Table 6A) in quantitative PCR experiments with blood-derived RNA. Expression of *PMS2* did not correlate with the lead SNP on chromosome 7 (Supplementary Table 6B). Despite these associations not reaching significance, the effect direction in our analyses and in GTEx was the same for the SNPs available at this portal.

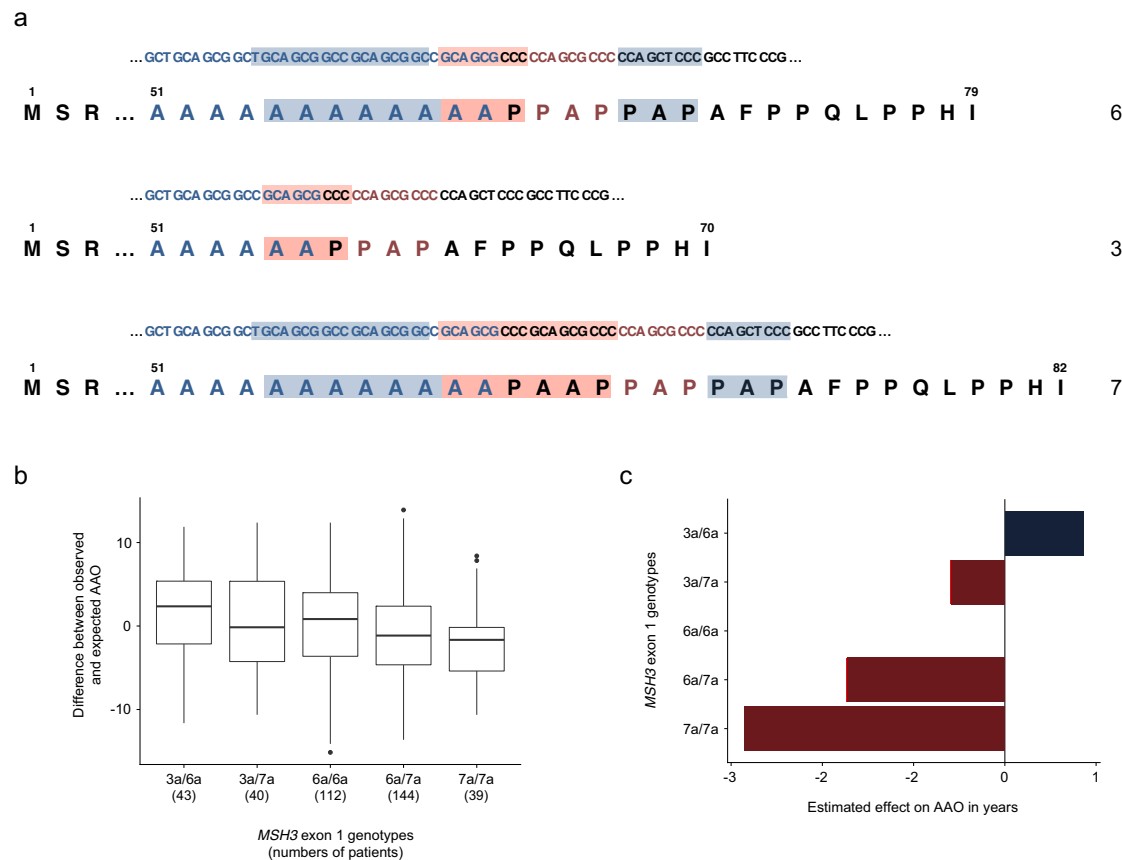

**Fig. 3 The length polymorphism in exon 1 of MSH3 and its correlation within different genotypes with age at onset (AAO) in X-linked dystonia-parkinsonism (XDP) patients. a** cDNA and amino-acid sequences showing the wild-type sequence, the shortest (c.162_179del;199_207del), and the longest (c.181_189dup) allele of *MSH3* exon 1, i.e., 6a, 3a, and 7a alleles, respectively. The sequences are based on the *MSH3* transcript ID ENST00000265081.7 (https://www.ensembl.org). Blue shading highlights the region deleted in the shortest allele (stretch of six alanines (A) and three other amino acids (proline, alanine, proline—PAP)). Red shading indicates the region duplicated in the longest allele. M: methionine (start codon); S: serine, R: arginine, AFPPQLPPHI: amino-acid sequence from the length polymorphism to the end of *MSH3* exon 1. **b** Difference between the observed and predicted (based on the hexanucleotide repeat number) AAO negatively correlates with the number of 7a alleles in a genotype. Box-plot elements: center line: median; box limits: upper and lower quartiles; whiskers: 1.5× interquartile range; points: outliers. **c** Disease onset is delayed by ~4 years in heterozygous carriers of the shortest allele (3a/6a) in comparison to the patients homozygous for the longest *MSH3* allele (7a/7a). Red color designates a harmful effect (earlier AAO), while blue depicts a protective effect (later AAO). The effect estimates are based on the linear regression model. Source data are provided as a Source data file.

## Discussion

Although penetrance of the XDP-causing mutation is widely considered to be complete, only at ~70 years of age all hemizygous *SVA* retrotransposon carriers will invariably manifest the phenotype. In our cohort, 94.5%, 57.0%, 19.2%, and 3.5% of individuals were not affected by XDP when they were 30, 40, 50, and 60 years old, respectively. Close to 50% of this AAO variability is due to the polymorphic repeat within the disease-causing *SVA* retrotransposon insertion[4,5]. Using a GWAS approach, we explored additional genetic modifiers of age-related disease penetrance in 353 unrelated XDP patients and identified three genome-wide significant signals. Combined with the influence of the hexanucleotide repeat in the XDP haplotype, these signals account for nearly two-thirds of the AAO variability in XDP.

Our study has several noteworthy and somewhat unexpected implications, the importance of which reaches beyond XDP. Although to date, over 50,000 unique associations between genetic variants and diseases or traits at genome-wide significance have been reported[12,13], they typically required large sample sizes (>1,000 participants) and had small effect sizes (often <10% of heritability/variability explained by the sum of several associated loci)[13–15]. Our finding is particularly interesting and encouraging,

as it demonstrates that significant modifiers of disease penetrance can be detected even in relatively small but carefully selected and homogeneous samples of mutation carriers. The significance of the *MSH3*-related locus—and the likely large effect that it exerts—are further supported by the near-significant effect that was observed in an even smaller-sized sample of HD patients (*n* = 218; TRACK-HD cohort)[16].

A specific analytical challenge was the high degree of genetic homology in our sample stemming from a small locale with close relatedness. To account for this, we estimated the cryptic relatedness, excluded samples that were related (first cousins or closer) and included principal components into the analysis to control for any further population stratification. After adjusting the significance level by Bonferroni correction, a relatively high number of samples were significantly associated with the AAO. Since most of the significant variants were imputed, all of them were in strong linkage disequilibrium with the three lead SNPs.

The strongest signal was found on chromosome 5 within the *MSH3* gene, where alternative alleles at two independent loci are associated with an earlier disease onset. On the other hand, the alternative allele of the chromosome 7 signal is correlated with the AAO increase in our patients. Interestingly, although

strictly speaking not within the associated region on chromosome 7, the *PMS2* gene is situated in close proximity to it. This is remarkable given that both *MSH3* and *PMS2* encode proteins involved in DNA mismatch repair (MMR), a cellular pathway responsible for the repair of mismatched base pairs in DNA. It has been hypothesized that the MMR process introduces instability in stretches of various, partially unwound repeat expansions in an attempt to amend the mismatch. Indeed, the downregulation[17] or reduced expression[18] of *MSH3* has been shown to reduce instability and expansion of tri- and hexanucleotide repeats. Absence or reduced expression of *PMS2* in mouse or cellular models provided contradictory results, i.e., reduction[19] and accumulation[20] of repeat expansions. Thus, the two genes that are either within (*MSH3*) or directly adjacent (*PMS2*) to the regions associated with AAO in XDP in our GWAS encode proteins that not only interact with each other and participate in the same cellular pathway, but their combined action has also been hypothesized to contribute to instability in stretches of nucleotide repeats similar to those within the XDP-causing insertion. Although the pathway analysis that we performed did not indicate any genetic pathway significantly associated with AAO at the stringent significance threshold that we used, this is not surprising, given our low sample size and consequently the relatively low power of this analysis.

Importantly, in 2017, *MSH3* was linked to HD disease progression in a human study that also employed a GWAS[16]. More recently, *MSH3* and *PMS2* were implicated in modifying AAO in HD through GWAS approaches, albeit in much larger sample sizes (>9,000 patients)[7]. HD is a severe movement disorder caused by trinucleotide repeat $(CAG)_n$ expansions in the coding region of the *HTT* gene resulting in an extended stretch of glutamine residues. Although the main presentation of HD is chorea, neuropathologically, HD and XDP both affect primarily the striatum and display loss of medium spiny neurons[21,22]. Both diseases typically affect patients in their early forties and pre-symptomatic neurodegeneration of the striatum is a phenomenon seen in both HD[21] and XDP[23,24]. Furthermore, clinical signs of HD may include dystonia and parkinsonism. Finally, HD and XDP are adult-onset neurodegenerative disorders precipitated/modified by nucleotide repeat expansions, while missense variants in both disease genes, *HTT* and *TAF1*, respectively, cause a childhood-onset neurodevelopmental syndrome with intellectual disability[25,26].

None of the SNPs in genes encoding other proteins relevant to MMR reached genome-wide significance in our study, despite the fact that some of them (i) showed signals of various strengths in the recent GWAS investigating association with AAO in >9,000 HD patients[7] or (ii) have been reported as associated with somatic expansion scores in blood DNA of >700 HD patients[27]. Possibly, our study was not powered to detect signals beyond the strongest associations that we were already able to identify. Alternatively, these other MMR genes may not play an important role in XDP, or the modifying alleles may be absent or much less common in the Filipino population.

Such population-specific effects apply to length polymorphisms in exon 1 of *MSH3*. Namely, while the frequency of the 6a allele (wild-type length) in a mixed-ethnicity HD patient population is comparable to that in our study (~58%[6] vs. 54%), the prevalence of the shortest (3a) and longest (7a) *MSH3* alleles differed considerably (~26% and ~10%[6] in HD vs. 10% and 32% in XDP). In the East Asian population, the frequency of the 7a allele also seems to be higher in comparison to the 3a allele (Supplementary Table 3). Furthermore, out of 16 different haplotypes previously detected in ethnically heterogeneous HD and myotonic dystrophy type 1 (DM1, OMIM # 160900) cohorts[6], only six were observed in our exclusively Filipino XDP

cohort of similar size (3a, 6a, 7a, 7b, 7c, and 7d, with the latter three arising from 7a and various combinations of two rare SNPs (rs2405875, p.Ala54Ala and rs2405877, p.Pro64Ala) in several patients). Nevertheless, the 3a allele is associated with delayed disease onset in HD (increasing AAO by 1 year), as is in our study (increasing AAO by 2 years)[6]. The same protective association was reported for HD progression and for AAO in DM1 patients[6], although this could not be shown in relatively small cohorts of spinocerebellar ataxia 3 (SCA3, OMIM # 109150) and Friedreich's ataxia (OMIM # 229300)[28]. In addition, we demonstrated that the longest (7a) *MSH3* allele correlated with an earlier disease onset (almost 3 years) in homozygous carriers compared to patients with wild-type *MSH3* length (6a). In a study of >100 HD patients, the 3a and 7a alleles were linked with lower and higher *MSH3* expression in comparison to 6a, respectively[6]. Our expression analyses using blood-derived RNA observed the same effect that, however, did not reach statistical significance, possibly because of low sample size (Supplementary Table 6A). The GTEx portal queries revealed an increase of *MSH3* and *PMS2* expression associated with alternative alleles in rs245013 and rs62456190, respectively, and a reduction of *MSH3* expression correlated with an alternative allele in rs1650697, showing the same effect expression as our own analyses (Supplementary Table 6). Of note, eQTL data imply that the loci we identified may explain a fraction of the expression differences in our genes of interest and serve as a guide as to which SNPs/genes to follow-up on in future functional studies. Out of >700 SNPs genotyped or imputed in our study and accessible through the GTEx portal, <50 showed genome-wide significance in our GWAS. Possible reasons for this relatively low number include (i) small minor allele frequency in the studied population (XDP patients) and (ii) limited sample size not allowing for the detection of small effects. On the other hand, SNPs not present as eQTLs in the GTEx, but identified through our GWAS, may tag a modifier acting through mechanisms other than a change of gene expression. For example, very recently, nuclear localization[29,30] and nuclear export[29] signals (NLSs and NLEs, respectively) have been identified within MSH3 as important for determining the levels of this protein in the nucleus and its ability to cross the nuclear envelope. Furthermore, NLS1 (encoded by *MSH3* exon 2) lies in close proximity to the length polymorphism in exon 1 and it has been shown that the shortest MSH3 (encoded by the 3a allele) is more prone to staying in the cytoplasm in comparison to the wildtype (6a) protein[29]. This is in agreement with the increase of AAO that is observed in XDP patients carrying the 3a allele, as the absence of MSH3 from nucleus would prevent it from introducing instability and would thus be protective.

From a personalized medicine point of view, our findings may help prioritize patients for clinical trials and may even serve as a basis for gene- or modifier-targeted therapeutic approaches. The recent randomized, double-blind, phase 1-2a trial with an antisense oligonucleotide in patients with early HD resulted in dose-dependent reductions in concentrations of mutant huntingtin[31]. Although it is conceivable that similar treatment options may be applicable to XDP, currently, it is unclear whether repeats with the *SVA* insertion are expressed or targetable. Thus, lowering MSH3 levels and thereby negatively influencing somatic instability (as was previously suggested for HD[32]) may be, at present, a more attainable treatment option for XDP.

In conclusion, our GWAS for genetic factors influencing age-related penetrance of XDP revealed strong signals within biologically relevant genes implying shared disease-modifying pathways with HD. More generally speaking, our findings may bear additional potential for a better understanding of other repeat expansion disorders.

## Methods

**Study participants and ethical review.** All individuals (patients and controls) investigated in the present study were of Filipino ethnic origin. Upon genome-wide SNP genotyping of DNA samples from 458 men with XDP, eight samples with low genotyping quality and samples from 97 patients related to the included individuals were removed before further analyses (Supplementary Table 1). Therefore, pairwise kinship coefficients were calculated for all samples and pairs with a kinship coefficient greater or equal 0.125 (first-cousin or closer) were defined as related. From each resulting family cluster, a single individual was selected based on the call rate on genotype level. Thus, we performed the GWAS in the remaining 353 participants (332 previously reported[3–5,33,34] and 21 newly analyzed). Samples from 37 more recently enrolled XDP patients and 162 healthy Filipino control individuals were investigated in post-GWAS genetic analyses. All participants gave informed consent and underwent a standardized neurological examination by movement disorder specialists. Participant enrollment and data analysis were approved by the Ethics Committees of the University of Lübeck, Germany, Massachusetts General Hospital, Boston, USA, Metropolitan Medical Center, Manila, Philippines, and Jose Reyes Medical Center, Manila, Philippines.

**Genetic and statistical analyses.** Genomic DNA of study participants was extracted from peripheral blood leukocytes. The presence of the *SVA* retrotransposon insertion in intron 32 of *TAF1* and the number of hexanucleotide repeats within this insertion was investigated by capillary-electrophoresis-based fragment analysis performed on the ABI 3500xL Genetic Analyzer (Applied Biosystems) and analyzed using GeneMapper software (Applied Biosystems). The fragments examined in this way were initially PCR amplified using primers and conditions listed in Supplementary Table 7[4]. To enable sizing of the repeats, one of the primers was labeled with a FAM tag.

DNA was subsequently analyzed by genome-wide SNP genotyping using the Infinium Global Screening Array with Custom Content (GSA; Illumina Inc.), which contains 645,896 variants. The 443,059 SNPs that passed quality control (PLINK1.9, SNP call rate >98%, minor allele frequency (MAF) > 1%, sample call rate >98%, deviation from mean heterozygosity ≤5 standard deviations) were imputed using SHAPEIT2[35] + IMPUTE2[36] with the public part of the HRC reference panel (release 1.1, The European Genome-phenome Archive, EGAS00001001710)[37] to a total of 39,083,699 variants. After filtering out all SNPs with a minor allele frequency lower than 5% or an imputation info score below 0.3, a total of 4,990,999 SNPs remained to be analyzed for association. As an additional quality control step, we checked our samples for deviations from the East Asian population (Supplementary Fig. 3).

rs1650697 was additionally genotyped by Sanger sequencing of *MSH3* exon 1 in 380 patients and 162 healthy Filipino controls (Supplementary Tables 1 and 7). We used gnomAD[38] to assess the frequencies of these variants globally, and in East and South Asian populations.

To evaluate the association and impact of different genetic variants on AAO in XDP, we employed a multi-step approach. Initially, we performed a GWAS on normalized AAO (i.e., centralized by mean and standardized by standard deviation) to identify candidate SNPs using a linear model, taking into account genotype uncertainty resulting from imputation (SNPTEST v2.5.1). We included the polymorphic $(CCCTCT)_n$ repeat number as a covariate in the model. Hence, the resulting regression estimates were adjusted for the repeat number. Additionally, we included the first 10 principal components into the model, to adjust for a potential population stratification. We considered SNPs with a p-value below $5.0 \times 10^{-8}$ as significant, to account for multiple testing in the GWAS context.

Afterward, we performed a pathway analysis (MAGMA v1.08)[39] aggregating the previously calculated p-values in 11,348 genes and 14,571 different genetic pathways[40]. Accounting for the number of genes and pathways, we considered genes with a p-value below $4.4 \times 10^{-6}$ (0.05/11348) and pathways with a p-value below $3.4 \times 10^{-6}$ as significant.

Next, significantly associated SNPs were reanalyzed using a conditional strategy (using R3.6)[41]. Therefore, we iteratively recalculated the regression coefficients of all SNPs per region adjusted for the SNP with the lowest p-value in this region, in order to determine whether one or several SNPs are responsible for the signal within a given chromosomal region.

Unsolicited sequence changes adjacent to rs1650697 were genotyped using Sanger sequencing and analyzed using linear models, again adjusting for the hexanucleotide repeat number to evaluate the association of these variants with AAO. If more than one single sequence change was found in a region, we estimated haplotypes including these variants using a standard expectation maximization algorithm and analyzed the influence of different haplotypes on AAO by linear regression, and adjusted for hexanucleotide repeat number. In cases where regions included in-frame indels, we additionally analyzed changes in length at the protein level by applying linear regression on AAO adjusted for hexanucleotide repeat number.

Since we observed an overlap between single-nucleotide polymorphism associated with AAO in HD[7] and variants detected by our GWAS, we constructed a PRS for AAO in HD using gender-specific effect estimates of the 21 lead SNPs from[7] and calculated this for all of our 353 XDP patients from the GWAS. With this, we calculated a linear regression model predicting AAO in XDP by the *SVA* repeat number and the PRS.

**Expression analyses.** We investigated the expression traits associated with variants of interest using the GTEx portal (gtexportal.org, Data Source: GTEx Analysis Release V8 (dbGaP Accession phs000424.v8.p2))[8], the BrainSeq dataset[9] from the eQTL Catalog[10], and the UKBEC dataset[11].

In addition, we analyzed *MSH3* and *PMS2* expression levels in blood via quantitative PCR (qPCR) analysis of cDNA synthesized from blood-derived RNA ($n = 40$). Real-time qPCR was performed using SYBR Green chemistry (Fermentas, Waltham, MA) on a Light Cycler 480 (Roche Applied Science). Expression levels of genes, measured in triplicates, were determined using the Advanced Relative Quantification method by the Roche LightCycler software (release 1.5.0)[13] and normalized against the expression of *GAPDH* and *YWHAZ*. Amplification conditions and primers are listed in Supplementary Table 7. The influence of investigated genotypes on corresponding expression levels was analyzed using linear models.

**Reporting summary.** Further information on research design is available in the Nature Research Reporting Summary linked to this article.

## Data availability

The authors declare that all data supporting the findings of this study are included in the article and its supplementary information files or are available from the corresponding authors upon reasonable request. Furthermore, the authors used the following databases: (i) The Haplotype Reference Consortium (HRC) reference panel (release 1.1, The European Genome-phenome Archive EGAD00001002729), (ii) The Genome Aggregation Database (gnomAD) (v2.1.1, https://gnomad.broadinstitute.org/gene/ENSG00000113318?dataset=gnomad_r2_1), (iii) The Genotype-Tissue Expression (GTEx) project (https://www.gtexportal.org/home/gene/MSH3#eQTLBlock, Analysis Release V8 (dbGaP Accession phs000424.v8.p2)), (iv) the BrainSeq dataset from the eQTL (expression quantitative trait locus) Catalog (https://www.ebi.ac.uk/eqtl/Studies/), (v) as well as the United Kingdom Brain Expression Consortium (UKBEC) (http://www.brainac.org/). The GWAS summary statistics (source data for Fig. 1) are available through the GWAS Catalog accession GCST90014263 (ftp://ftp.ebi.ac.uk/pub/databases/gwas/summary_statistics/GCST90014263). Source data are provided with this paper.

## Code availability

The authors declare that the codes used in this study are available from the corresponding author upon reasonable request.

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

## Acknowledgements
The study was supported by the Deutsche Forschungsgemeinschaft (DFG; FOR 2488 to C.K., N.B., K.L., F.J.K, I.R.K, and A.W.; SFB 936 [project C5] to C.K.), the Collaborative Center for X-linked Dystonia-Parkinsonism at Massachusetts General Hospital (to N.B., C.K., L.J.O., N.S., T.M.-B., and C.G.), the Else Kröner Fresenius Foundation (to N.B.), intramural funds from the University of Lübeck (to C.K. and M.D.-M.), and a career development award from the Hermann and Lilly Schilling Foundation (to C.K.). Furthermore, this work received infrastructure support by the DFG Cluster of Excellence "Precision Medicine in Chronic Inflammation" (PMI, EXC2167 to A.F.). We thank the XDP patients and their relatives for their participation in this study; the Sunshine Care Foundation and members of their staff (including genetic councilors N.G.M. Ganza and B.B. Lagarde) for the infrastructure they have established and for their devoted and hard work; Dr. L.V. Lee, Dr. R. Kaji, and Dr. T. Kawarai for referral of DNA samples and clinical information; Dr. L. Bertram who was involved in early scientific discussions of the study, acquisition of funding, and supervision of the collection of the GSA-based genotyping data; and H. Pawlack for technical assistance, including genotyping of DNA samples.

## Author contributions
B.-H.L. performed the genome-wide association analysis and data interpretation and co-drafted the manuscript; C.K. participated in conception and design, collected and interpreted data, and significantly contributed to the writing of the manuscript and later its critical revision; J.P. did the sequencing, genotyping, and expression analyses and critically revised the manuscript; A.D. interpreted data and critically revised the manuscript; N.B. collected and analyzed clinical data and critically revised the manuscript; K.G. interpreted data and critically revised the manuscript; R.L.R., R.D.J., G.S., and C.C.E.D. collected and interpreted data and critically revised the manuscript; M.W. performed array genotyping (GSA) for the discovery study and critically revised the manuscript; S.S. and M.D.-M. interpreted data and critically revised the manuscript; J.Q. and P.O. performed sequencing/genotyping and critically revised the manuscript; P.A., C.G., N.S., T.M.-B., U.M., and H.H. collected data and critically revised the manuscript; F.K., A.R., and P.B. did the genotyping analysis and critically revised the manuscript; A.F. performed array genotyping (GSA) for the discovery study and critically revised the manuscript; V.D. collected data and critically revised the manuscript; K.L. participated in conception and design and critically revised the manuscript; L.J.O. performed genotyping and the haplotype analysis, interpreted data and critically revised the manuscript; F.J.K. participated in conception and design, did the expression analyses, interpreted data, and critically revised the manuscript; I.R.K. participated in conception and design, did the genome-wide association analysis, interpreted data, and critically revised the manuscript; A.W. participated in conception and design, analyzed and interpreted data, and co-drafted the manuscript. All authors gave their final approval of the version to be published and agreed to be accountable for all aspects of the work.

## Funding

## Competing interests
The authors declare no competing interests.
