## [Peer Review File · Nature Communications]

Reviewers' Comments:

Reviewer #1:

Remarks to the Author:

This very interesting MS examines genetic modifiers of age at onset in a cohort of X-linked dystonia-parkinsonism males. The disease is caused by an intronic hexanucleotide repeat expansion in intron 32 of the TAF1 gene, and age at onset is modified by the length of the repeat locus. As this accounts for around half of the variance in age at onset the authors hypothesised that the age of onset of disease would also be modified by variants in other genes and they conducted a genome-wide association study in a cohort of 353 subjects with the disease.

The data they report shows that there are three GW loci – two on chromosome 5 close to the MSH3/DHFR locus and one on chromosome 7 in or close to the EIF2AK1, ANKRD61 and PMS2 genes. These data are indeed very interesting, but the paper would benefit from some further analyses and some adjustments to the context in the introduction and discussion.

Major comments:

1. In the title and the abstract the authors emphasise the relevance of the findings in their study to HD and by implication other repeat disorders. This is true but it has been observed previously so is less novel than the authors propose (PMID: 27044000, PMID: 31607598). Both the title and the abstract should be amended to reflect this.
2. The authors comment that these data will help in clinical diagnostics and prognostics. They need to be much more explicit about how these findings might help patients. Predicting changes in age of onset in individuals from the effects in populations is challenging and it is hard to use this information in clinical settings currently, even in HD.
3. The authors make no comment about the possibility of common therapeutic developments if as they propose, somatic instability of repeats underlies the manifestation of disease in both HD and XDP.
4. The authors mention that it is notable that the three loci implicate MSH3 and PMS2, both involved in mismatch repair – however the likelihood of this occurring is not quantified in any way. A pathway analysis of the full GWAS data, using readily available softwares such as MAGMA, could supply this and should be performed.
5. The authors have sensibly taken a conservative threshold of SNPs with frequencies of >5% for their GWAS, especially given its small size. However, it would be interesting to see the results of using different thresholds, although interpretation of these data would need to be circumspect. In particular it might reveal further potential associations and pathway analyses of these data might provide more biological insights.
6. The authors mention the HD GWAS (17) and the papers from Hensman-Moss et al. (16) and Flower et al. (5) that implicate MSH3 and PMS2 as significantly associated loci of interest in the onset and progression of HD. However, they do not cite Ciosi et al. (PMID: 31607598) that connects somatic expansion in myotonic dystrophy and HD with the variants in MSH3, as well as other DNA repair genes.
7. In order to show the relationship with variants in HD it would be useful to construct a polygenic age at onset score from the HD GWAS data presented in ref 17 (the sum of the number of minor alleles at each locus weighted by their effect size in the GWAS in GeM-HD) and investigate how much of the variance in age at onset in the XDP GWAS can be accounted for by this. This would reveal whether there were other influences outside the genome-wide significant influences. Again this could be run at different SNP MAF thresholds.
8. The authors have explored the effects of the variants in GTex (line 199 onwards). Most of the significances cited are fairly marginal, except for that in whole blood (line 208) where the minor allele appears to confer a substantial decrease in expression. The authors conducted their own study in a small sample of 35 subjects (Lines 213-214) – what is the power of such a sample size to see the effect seen in the GTex data? Ideally these data should be presented for assessment by the reader, and a comment made on their ability to confirm or refute the GTex data. It would greatly strengthen

the MS if further samples could be analysed to investigate this interesting finding further.

9. Lines 235-236. The authors talk about loci accelerating and delaying disease onset. This is misleading as all variant loci have two alleles (often referred to as reference and minor alleles). This means that at any locus associated with age at onset of disease there will be an allele accelerating onset and one delaying it. The wording in the MS needs amending to reflect this.

Minor comments

10. OMIM numbers should be given for each inherited disease on first mention.

11. In Figure 3B and the text referring to it the alleles are mentioned as coding changes. They do indeed change the encoded protein but it remains unclear whether the coding changes or the changes in the DNA/RNA are important in modifying age at onset in repeat disorders, so the figure and accompanying text need amending to reflect this.

12. The relationship of the variants in MSH3 detected in reference 5 to those detected in this MS could be made much clearer for the reader. The relationship of these variants to direction of expression of MSH could also be made clearer.

13. Line 228-30 – this sentence seems to have a word missing as it doesn't make sense right now.

14. Line 267. Ref 17 used >9000 subjects.

Reviewer #2:

Remarks to the Author:

In this report the authors perform GWAS in a small population of patients with X-linked dystonia-parkinsonism to search for genetic modifiers to explain variability in age of onset. The paper is overall well-written and poses an interesting question facing a number of genetic conditions regarding how detailed genomic analysis might personalize care.

The manuscript would benefit from additional discussion of statistical power and analytic methods. The population under study is from a small locale with a population of roughly 4 million with a likely high degree of genetic homology. Given the small sample size the observance of such a large number significant SNPs seems surprising and the frequency of these SNPs in this Filipino population is not discussed nor how this relates to the power of this study. Filtration was presumably not based on the frequencies within this same population. Furthermore, it is unclear if multivariate correction for the p-values was performed and with what method as only a specific p-value for significance is given.

The meaning of eQTL data is unclear. It is unlikely these SNPs are the only notable eQTLs in the genes of interest within the population and the authors do not state how these particular SNPs might rank amongst the others or comment on why some significant eQTLs might be identified by GWAS yet others may not.

Finally, the discussion is overly detailed and entirely speculative. Experimental studies are required to assess the detailed hypotheses presented. There is no clear mechanistic evidence or data to support the suggestion that there is connection to prior HD genetic modification studies which, at this stage, is entirely observational.

As a minor point, Figure 3A is unnecessary and the genotype information could be added to panel 3B.

Reviewer #3:

Remarks to the Author:

The authors assessed genetic modifiers of the age at onset of X-linked dystonia-parkinsonism. An inverse correlation of repeat numbers and age of onset has previously been described.

A genome-wide association study in > 300 X-linked dystonia-parkinsonism cases and healthy controls

identified associated SNPs in genomic regions on chromosome 5 and chromosome 7.

The study is thoroughly conducted, well written and presents comprehensible conclusions.

Remarks to the authors:

The authors performed a GWAS in a Filipino population. The majority of the population is ethnically mixed as the population has been influenced by the Spanish occupiers and immigration from Taiwan and from southern China. The authors mention that 37 XDP patients and 162 healthy Filipino controls had been included in a post-GWAS genetic analysis. It is unclear which controls (ethnic groups) had been included in the first part of this association study. Please describe in detail the population structure of this GWAS (cases and controls) and the statistical approach used to address genetic diversity and population structures.

Expression quantitative trait locus query had been performed for relevant SNPs. Only for some SNPs information on tissue-dependent expression seemed to be available. I recommend consulting other databases (UK Brain Expression Consortium, BRAINEAC ...) and to summarize the results for each associated SNP in a table. If no information for important SNPs (e.g rs33003) has been deposited the authors should consider expression analyses (quantitative PCR) from XDP brain tissue which is available to the authors.

Re: Manuscript no: NCOMMS-20-38460A “Identifying novel genetic modifiers of age-associated penetrance in X-linked dystonia-parkinsonism”

We would like to thank to the anonymous referees and the editors for reviewing our above-mentioned manuscript and for the very helpful suggestions.

We are responding to the recommendations of the reviewers in detail as follows:

Reviewer #1

1. In the title and the abstract the authors emphasise the relevance of the findings in their study to HD and by implication other repeat disorders. This is true but it has been observed previously so is less novel than the authors propose (PMID: 27044000, PMID: 31607598). Both the title and the abstract should be amended to reflect this.

Response: To follow the reviewer’s advice, we have now omitted the second part of our initial title. The new title reads: “Identifying novel genetic modifiers of age-associated penetrance in X-linked dystonia-parkinsonism”. Similarly, we have toned down the abstract by removing the last sentence.

2. The authors comment that these data will help in clinical diagnostics and prognostics. They need to be much more explicit about how these findings might help patients. Predicting changes in age of onset in individuals from the effects in populations is challenging and it is hard to use this information in clinical settings currently, even in HD.

Response: The reviewer is right, and we have now removed the statement in question (“our findings will impact on specific counseling of mutation carriers based on their modifier profile,”) from the Discussion.

3. The authors make no comment about the possibility of common therapeutic developments if as they propose, somatic instability of repeats underlies the manifestation of disease in both HD and XDP.

Response: We thank the reviewer for pointing this out. We have now expanded on this and also included a recent reference, so that our comment on the therapeutic approaches is rephrased from: “The recent randomized, double-blind, phase 1-2a trial with an antisense oligonucleotide in patients with early HD resulted in dose-dependent reductions in concentrations of mutant huntingtin and it is conceivable that similar treatment options may be amenable to XDP.” to “The recent randomized, double-blind, phase 1-2a trial with an antisense oligonucleotide in patients with early HD resulted in dose-dependent reductions in concentrations of mutant huntingtin. Although it is conceivable that similar treatment options may be applicable to XDP, currently, it is unclear whether repeats with the SVA insertion are expressed or targetable. Thus, lowering MSH3 levels and thereby negatively influencing somatic instability (as was previously suggested for HD (PMID: 30844400)) may be, at present, a more attainable treatment option for XDP.”.

4. The authors mention that it is notable that the three loci implicate MSH3 and PMS2, both involved in mismatch repair – however the likelihood of this occurring is not quantified in any way. A pathway analysis of the full GWAS data, using readily available softwares such as MAGMA, could supply this and should be performed.

Response: We thank the reviewer for this suggestion. We followed the advice and used MAGMA to gain further insight into our findings, showing that *MSH3* is significantly associated with AAO. Given our limited sample size, we were, however, not surprised that we were not able to identify a significantly associated pathway after adequate correction for multiple testing. We have now added this to the Methods section (“Afterwards, we performed a pathway analysis (MAGMA v1.08) aggregating the previously calculated p-values in 11,348 genes and 14,571 different genetic pathways. Accounting for the number of genes and pathways, we considered genes with a p-value below 4.4×10^{-6} ($0.05/11348$) and pathways with a p-value below 3.4×10^{-6} as significant.”), and the Results section (“When aggregating the p-values of single SNPs per gene, *MSH3* was significantly associated with AAO ($p=1.72 \times 10^{-12}$). We did not observe any genetic pathway to be significantly associated with AAO at the stringent significance threshold.”) of the manuscript.

5. *The authors have sensibly taken a conservative threshold of SNPs with frequencies of >5% for their GWAS, especially given its small size. However, it would be interesting to see the results of using different thresholds, although interpretation of these data would need to be circumspect. In particular it might reveal further potential associations and pathway analyses of these data might provide more biological insights.*

Response: We agree with the reviewer and indeed explored associations of SNPs with frequencies of 1% but detected no further hits. However, at a MAF of 1%, the absolute frequencies of minor alleles of these SNPs are expected to be only 3.53 in our limited sample, meaning that no homozygotes and only very few heterozygotes are to be expected among the patients. We would therefore prefer not to include these analyses in the manuscript.

6. *The authors mention the HD GWAS (17) and the papers from Hensman-Moss et al. (16) and Flower et al. (5) that implicate *MSH3* and *PMS2* as significantly associated loci of interest in the onset and progression of HD. However, they do not cite Ciosi et al. (PMID: 31607598) that connects somatic expansion in myotonic dystrophy and HD with the variants in *MSH3*, as well as other DNA repair genes.*

Response: We apologize for overlooking this relevant reference and have now added it to the manuscript (“None of the SNPs in genes encoding proteins relevant to MMR, reached genome-wide significance in our study, despite the fact that some of them i) showed signals of various strengths in the recent GWAS investigating association with AAO in >9,000 HD patients (PMID: 31398342) or ii) have been reported as associated with somatic expansion scores in blood DNA of >700 HD patients (PMID: 31607598).”).

7. *In order to show the relationship with variants in HD it would be useful to construct a polygenic age at onset score from the HD GWAS data presented in ref 17 (the sum of the number of minor alleles at each locus weighted by their effect size in the GWAS in GeM-HD) and investigate how much of the variance in age at onset in the XDP GWAS can be accounted for by this. This would reveal whether there were other influences outside the genome-wide significant influences. Again this could be run at different SNP MAF thresholds.*

Response: We appreciate this helpful suggestion and computed the suggested polygenic score in our XDP patients. Interestingly, this accounted for about 10% of the variance in AAO, compared with the about 25% explained by the three lead SNPs of our GWAS. We added this information to the Methods section with: “Since we observed an overlap between single nucleotide polymorphism associated with AAO in HD (PMID: 31398342) and variants

detected by our GWAS, we constructed a polygenic risk score for AAO in HD using gender-specific effect estimates of the 21 lead SNPs from PMID: 31398342 and calculated this for all of our 353 XDP patients from the GWAS. With this we calculated a linear regression model predicting AAO in XDP by the SVA repeat number and the PRS.” To the Results section, we added: “We constructed a polygenic risk score (PRS) according to PMID: 31398342 to predict the AAO in XDP and we observed an increase of 1.30 years ($p = 1.06 \times 10^{-9}$) for every increase of one in the PRS. In total, the PRS explained about 10% of the remaining variance in AAO, while the three lead SNPs of our GWAS explained more the 25% of the remaining variance. Overall, the PRS for HD was constructed based on a GWAS performed in observations from a different genetic population, thus the effect estimates are not necessarily transferable to a Filipino population.”.

8. The authors have explored the effects of the variants in GTex (line 199 onwards). Most of the significances cited are fairly marginal, except for that in whole blood (line 208) where the minor allele appears to confer a substantial decrease in expression. The authors conducted their own study in a small sample of 35 subjects (Lines 213-214) – what is the power of such a sample size to see the effect seen in the GTex data? Ideally these data should be presented for assessment by the reader, and a comment made on their ability to confirm or refute the GTex data. It would greatly strengthen the MS if further samples could be analysed to investigate this interesting finding further.

Response: We absolutely agree with the reviewer on this and strongly feel that showing and discussing the requested data would enhance the manuscript. We have in the meantime collected RNAs from five more patients and have performed *PMS2* expression analyses, as well. Under the current circumstances and traveling restrictions, it is not likely that we will collect many more RNA samples to increase the power considerably. However, following the reviewer’s suggestion, we now present the concrete data in Supplementary Table 8. Indeed, as the reviewer probably suspected, the effect direction in our and the GTex data is the same. We have now adjusted the Results and Discussion sections accordingly (“*MSH3* expression was not associated with any of the lead SNPs on chromosome 5 nor with variants in exon 1 of *MSH3* at a significance level of $0.05/7=0.0071$ (Supplementary table 8A) in quantitative PCR experiments with blood-derived RNA. Expression of *PMS2* did not correlate with the lead SNP on chromosome 7 (Supplementary table 8B). Despite these associations not reaching significance, the effect direction in our analyses and in GTex was the same for the SNPs available at this portal.” and “Our expression analyses using blood-derived RNA observed the same effect that, however, did not reach statistical significance, possibly because of low sample size (Supplementary Table 8A).”).

9. Lines 235-236. The authors talk about loci accelerating and delaying disease onset. This is misleading as all variant loci have two alleles (often referred to as reference and minor alleles). This means that at any locus associated with age at onset of disease there will be an allele accelerating onset and one delaying it. The wording in the MS needs amending to reflect this.

Response: We thank the reviewer for reading our manuscript so carefully and drawing our attention to this inaccuracy. We have now amended the wording in the manuscript (“The strongest signal was found on chromosome 5 within the *MSH3* gene where alternative alleles at two independent loci are associated with an earlier disease onset. On the other hand, the alternative allele of the chromosome 7 signal is correlated with the AAO increase in our patients.”).

10. OMIM numbers should be given for each inherited disease on first mention.

Response: We have now added the respective OMIM numbers upon first mentioning of each disease.

11. In Figure 3B and the text referring to it the alleles are mentioned as coding changes. They do indeed change the encoded protein but it remains unclear whether the coding changes or the changes in the DNA/RNA are important in modifying age at onset in repeat disorders, so the figure and accompanying text need amending to reflect this.

Response: The reviewer is correct – at present, we do not know whether these variants act at the DNA/RNA level, or at the protein level, or at both or at neither of those levels. It is conceivable that the protective alleles act through reducing the expression of *MSH3* or by reducing the levels of the *MSH3* protein in the nucleus (PMID: 32284349).

Thus, we no longer state in the manuscript that any of the length polymorphisms “confer protection” or “have an affect” on AAO, but rather that they are “associated” or “correlated” with delayed disease onset.

In the Discussion, we elaborate: “In a study of >100 HD patients, the 3a and 7a alleles were linked with lower and higher *MSH3* expression in comparison to 6a, respectively.” and “Apart from influencing gene expression, modifiers of AAO in XDP may act through a different mechanism. Namely, very recently nuclear localization and nuclear export signals (NLSs and NLEs, respectively) have been identified within *MSH3* as important for determining the levels of this protein in the nucleolus and cytoplasm and its ability to cross the nuclear envelope. Furthermore, NLS1 (encoded by *MSH3* exon 2) is in close proximity to the length polymorphism in exon 1 and it has been shown that the shortest *MSH3* (encoded by the 3a allele) is more prone to stay in the cytoplasm in comparison to the wild type (6a) protein. This is in agreement with the increase of AAO that is observed in XDP patients carrying the 3a allele, as the absence of *MSH3* from nucleolus would prevent it from introducing instability and would thus be protective.”.

12. The relationship of the variants in *MSH3* detected in reference 5 to those detected in this MS could be made much clearer for the reader. The relationship of these variants to direction of expression of *MSH* could also be made clearer.

Response: In order to make the relationship between the *MSH3* variants that we detected and those previously reported by Flower et al., 2019 clear, we adapted the nomenclature from this publication (i.e., 6a, 3a, and 7a designation of the length polymorphism: “Of note, the absence or presence of three in-frame sequence length polymorphisms [c.162_179del (p.Ala57_Ala62del), c.199_207del (p.Pro67_Pro69del), and c.181_189dup (p.Ala61_Pro63dup) the first two of which were always detected together; Supplementary Table 3] form alleles of three different sizes: i) wildtype *MSH3*, ii) a 27-nucleotide/9-amino-acid shorter, and iii) a nine-nucleotide/3-amino-acid longer form previously described as 6a, 3a, and 7a, respectively (Figure 3A).⁵”).

For the relationship of those variants to direction of *MSH3* expression we now state in the Discussion: “In a study of >100 HD patients, the 3a and 7a alleles were linked with lower and higher *MSH3* expression in comparison to 6a, respectively.⁵ Our expression analyses using blood-derived RNA observed the same effect that however did not reach statistical significance, possibly because of low sample size (Supplementary Table 8A).”

Finally, we have now added our expression analyses results as Supplementary table 8.

13. Line 228-30 – this sentence seems to have a word missing as it doesn't make sense right now.

Response: Indeed, the word “they” was missing and this has now been corrected.

14. Line 267. Ref 17 used >9000 subjects.

Response: We have now amended this typo.

Reviewer #2

The manuscript would benefit from additional discussion of statistical power and analytic methods. The population under study is from a small locale with a population of roughly 4 million with a likely high degree of genetic homology. Given the small sample size the observance of such a large number significant SNPs seems surprising and the frequency of these SNPs in this Filipino population is not discussed nor how this relates to the power of this study. Filtration was presumably not based on the frequencies within this same population. Furthermore, it is unclear if multivariate correction for the p-values was performed and with what method as only a specific p-value for significance is given.

Response: We thank the reviewer for raising these points and apologize for not having been clear enough. Given that our study population indeed is likely to have a high degree of homology, we estimated cryptic relatedness from the genetic data and (relatively strictly) excluded samples who had an estimated relationship of cousins or closer (kinship coefficient ≥ 0.125). In addition, we included the first ten principal components calculated on the genetic data of the included samples to account for any further population stratification in our study population. Since we used an imputed data set, most of the significant SNPs are in relatively strong linkage disequilibrium, so that they are all correlated to one of the three lead SNPs (rs245013, rs33003 and rs62456190). The frequencies of these three SNPs are now given in Supplementary Table 3. After imputation we excluded all variants with an imputation info score below 0.3 and a minor allele frequency (in the Filipino study population) below 0.05. For the GWAS we included not only the first ten principal components but also the repeat number (in the SVA repeat expansion) to the model, so that the effect estimates for each SNP and therefore the corresponding p-values are already adjusted for these independent variables. To account for multiple testing, we adjusted the global significance threshold to 5×10^{-8} .

The meaning of eQTL data is unclear. It is unlikely these SNPs are the only notable eQTLs in the genes of interest within the population and the authors do not state how these particular SNPs might rank amongst the others or comment on why some significant eQTLs might be identified by GWAS yet others may not.

Response: Available eQTL data imply that the loci we identified likely explain a fraction of the expression differences in our genes of interest; in the absence of functional experiments, they serve as a guide as to which SNPs/genes to follow up on for functional studies. In the GTEx portal there are 1,375 eQTLs for *MSH3*, 742 of which were included (i.e., genotyped or imputed) in our GWAS. Our GWAS identified 60 genome-wide significant SNPs on chromosome 5. Of these 60, 47 are present in the GTEx portal and 41 are eQTLs for *MSH3* in brain tissue. Given that XDP is an exclusively neurologic disorder, we felt that this filtering based on tissue was helpful in prioritizing and justified. The possible reasons for identifying only 47 out 742 SNPs in our GWAS include: i) small minor allele frequency in the studied

population (XDP patients), and ii) small sample size not allowing for the recognition of the small effects.

On the other hand, SNPs not present as eQTLs in the GTEx but identified through our GWAS may be tagging a modifier that does not act through change of expression but other mechanisms.

Finally, the discussion is overly detailed and entirely speculative. Experimental studies are required to assess the detailed hypotheses presented. There is no clear mechanistic evidence or data to support the suggestion that there is connection to prior HD genetic modification studies which, at this stage, is entirely observational.

Response: We have now shortened the Discussion. Indeed, our findings warrant further experimental studies that are outside of the scope of the current manuscript. Nevertheless, we feel that even without currently available mechanistic evidence, our suggestion that there is a connection between our and prior HD genetic modification studies is more than plausible. Namely, the likelihood that two GWA studies performed on patients with two repeat expansion disorders indicate the involvement of the same proteins and conceivable mechanisms (mismatch DNA repair) solely by chance is very low.

As a minor point, Figure 3A is unnecessary and the genotype information could be added to panel 3B.

Response: We have now represented the information from Figures 3A and 3B in the new Figure 3A.

Reviewer #3

The authors performed a GWAS in a Filipino population. The majority of the population is ethnically mixed as the population has been influenced by the Spanish occupiers and immigration from Taiwan and from southern China. The authors mention that 37 XDP patients and 162 healthy Filipino controls had been included in a post-GWAS genetic analysis. It is unclear which controls (ethnic groups) had been included in the first part of this association study. Please describe in detail the population structure of this GWAS (cases and controls) and the statistical approach used to address genetic diversity and population structures.

Response: All individuals investigated in our study were of Filipino ethnic origin. We now state this more clearly in the first sentence of the Methods section (“All individuals (patients and controls) investigated in the present study were of Filipino ethnic origin.”). The GWAS study was performed only in XDP patients, i.e., no data from controls was used. We now clearly state this in the second sentence of the Methods (“Upon genome-wide single-nucleotide polymorphism (SNP) genotyping of DNA samples from 458 men with XDP,...”). As an additional quality control step, we checked our samples for deviations from the East Asian population (Supplementary Figure 1). The handling of population structure is described in the Methods section “Additionally, we included the first 10 principal components into the model, to adjust for a potential population stratification. We considered SNPs with a p-value below 5.0×10^{-8} as significant, to account for multiple testing in the GWAS context.”.

Expression quantitative trait locus query had been performed for relevant SNPs. Only for some SNPs information on tissue-dependent expression seemed to be available. I

recommend consulting other databases (UK Brain Expression Consortium, BRAINEAC ...) and to summarize the results for each associated SNP in a table. If no information for important SNPs (e.g rs33003) has been deposited the authors should consider expression analyses (quantitative PCR) from XDP brain tissue which is available to the authors.

Response: This is an excellent suggestion that we now implemented by adding four tables to our Supplementary Material. Supplementary Tables 5-7 show the results of our queries of the GTEx, eQTL Catalogue, and the UK Brain Expression Consortium for four relevant SNPs (three lead SNPs, rs245013, rs33003, rs62456190, and the coding SNP rs1650697 in *MSH3*). Given that these eQTL data i) used a variety of expression profiling technologies, including different quality control and statistical approaches and ii) interrogate different tissues in different numbers of patients, we preferred to display the results of our queries in separate tables as we felt that combining them in a single table would be misleading. Supplementary Table 8 contains our own expression analysis of the relevant polymorphisms in the blood-derived RNAs of 40 XDP patients. As only two postmortem brains are available to us, we believe that we could not perform statistically meaningful expression analyses.

Reviewers' Comments:

Reviewer #1:

Remarks to the Author:

The MS is much improved by the changes made after review. There are some minor points:

1. final sentence of the abstract "influencing" might be better than "conferring".
2. Line 246 onwards - although their study does show a genome-wide effect in a small population, the study by Hensman-Moss et al (ref 20) found a modifying effect in MSH3 in a similarly small population. So perhaps this is less remarkable - it might mean that this locus exerts a large effect and that is why it is detected. The discussion should be amended here.

Lesley Jones

Reviewer #2:

Remarks to the Author:

I thank the authors for their detailed responses addressing my comments regarding this manuscript. The revised Discussion is much improved. I also appreciated the response regarding concerns regarding how statistical power was achieved in the GWAS, it is now much clearer, however I did not see any modification to the text reflecting these clarifications. Similarly, while I again appreciated the discussion regarding the eQTLs, I was unable to find these clarifications in the revised manuscript either in the methods or the discussion.

Reviewer #3:

Remarks to the Author:

The authors have sufficiently answered my questions.

Re: Manuscript no: NCOMMS-20-38460A “Identifying novel genetic modifiers of age-associated penetrance in X-linked dystonia-parkinsonism”

We would like to thank the anonymous referees and the editors for reviewing our above-mentioned manuscript and for the constructive suggestions.

We are responding to the recommendations of the reviewers in detail as follows:

Reviewer #1

1. *Final sentence of the abstract "influencing" might be better than "conferring".*

Response: We have now replaced the word “conferring” with the word “influencing” in the abstract.

2. *Line 246 onwards - although their study does show a genome-wide effect in a small population, the study by Hensman-Moss et al (ref 20) found a modifying effect in MSH3 in a similarly small population. So perhaps this is less remarkable - it might mean that this locus exerts a large effect and that is why it is detected. The discussion should be amended here.*

Response: Thank you very much for drawing our attention to this point. Indeed, it might be that the effect of the *MSH3* locus is large, and this may be the reason why it was identified in our study. The argument that we are trying to make, that significant (and large-effect) modifiers can be detected even in small and carefully chosen cohorts, is thus even confirmed by the finding of Hensman-Moss and colleagues. Also, we identified another locus (on chromosome 7) that showed genome-wide significance. We thus modified the discussion as suggested by adding the following sentence: “The significance of the *MSH3*-related locus – and the likely large effect that it exerts – are further supported by the near-significant effect that was observed in an even smaller-size sample of HD patients (n=218; TRACK-HD cohort).” to the paragraph in question.

Reviewer #2

I thank the authors for their detailed responses addressing my comments regarding this manuscript. The revised Discussion is much improved. I also appreciated the response regarding concerns regarding how statistical power was achieved in the GWAS, it is now much clearer, however I did not see any modification to the text reflecting these clarifications.

Response: Thank you for this comment. We have now added the following comment to the discussion: “A specific analytical challenge was the high degree of genetic homology in our sample stemming from a small locale with close relatedness. To account for this, we estimated the cryptic relatedness, excluded samples that were more closely related than cousins and included principal components into the analysis to control for any further population stratification. After adjusting the significance level by Bonferroni correction, a relatively high number of samples were significantly associated with the AAO. Since most of the significant variants were imputed, all of them were in strong linkage disequilibrium with the three lead SNPs.”.

Similarly, while I again appreciated the discussion regarding the eQTLs, I was unable to find these clarifications in the revised manuscript either in the methods or the discussion.

Response: We have now included a more detailed discussion of the meaning of the eQTL data in general and with respect to our study: “Of note, eQTL data imply that the loci we

identified may explain a fraction of the expression differences in our genes of interest and serve as a guide as to which SNPs/genes to follow up on in future functional studies. Out of >700 SNPs genotyped or imputed in our study and accessible through the GTEx portal, <50 showed genome-wide significance in our GWAS. Possible reasons for this relatively low number include: i) small minor allele frequency in the studied population (XDP patients) and ii) limited sample size not allowing for the detection of small effects. On the other hand, SNPs not present as eQTLs in the GTEx, but identified through our GWAS, may tag a modifier acting through mechanisms other than a change of gene expression.”